# Spatiotemporal dynamics of information encoding revealed in orbitofrontal high-gamma

Erin L. Rich[1,2] & Joni D. Wallis[1,3]

High-gamma signals mirror the tuning and temporal profiles of neurons near a recording electrode in sensory and motor areas. These frequencies appear to aggregate local neuronal activity, but it is unclear how this relationship affects information encoding in high-gamma activity (HGA) in cortical areas where neurons are heterogeneous in selectivity and temporal responses, and are not functionally clustered. Here we report that populations of neurons and HGA recorded from the orbitofrontal cortex (OFC) encode similar information, although there is little correspondence between signals recorded by the same electrode. HGA appears to aggregate heterogeneous neuron activity, such that the spiking of a single cell corresponds to only small increases in HGA. Interestingly, large-scale spatiotemporal dynamics are revealed in HGA, but less apparent in the population of single neurons. Overall, HGA is closely related to neuron activity in OFC, and provides a unique means of studying large-scale spatiotemporal dynamics of information processing.

[1] Helen Wills Neuroscience Institute, University of California at Berkeley, Berkeley, CA 94720, USA. [2] Friedman Brain Institute and Department of Neuroscience, Icahn School of Medicine at Mount Sinai, New York, NY 10029, USA. [3] Department of Psychology, University of California at Berkeley, Berkeley, CA 94720, USA. Correspondence and requests for materials should be addressed to E.L.R. (email: erin.rich@mssm.edu)

In order for a basic understanding of brain function to emerge across disciplines, we must determine how the activity of single neurons relates to larger voltage fluctuations recorded as local field potentials (LFPs) or electrocorticographic (ECoG) potentials. These mesoscale signals are a valuable source of information: compared to single unit activity, they are easier to obtain and maintain over time, particularly in human subjects. Their accessibility allows us to draw comparisons between human and animal studies[1, 2], and the chronicity makes mesoscale potentials a promising input for brain machine interfaces[3–6]. In addition, devices recording these signals, particularly ECoG, can cover more cortical territory than single neuron recordings, allowing a larger-scale perspective of neurophysiology[7]. Therefore, understanding the information carried in these signals is critical to both basic and translational neuroscience.

Previous studies demonstrate strong links between neuron spiking and amplitudes of high-frequency LFPs/ECoG[8–13]. Not only are there temporal and trial-wise correlations[1, 13], but tuning properties observed in the local neuron population, typically a cortical column, align with those of the LFP at frequencies above ~60–80 Hz[6, 8]. These findings have led to the suggestion that high-frequency signals reflect aggregate activity of a local population of neurons, and could be used as a proxy for neuronal responses[14, 15]. However, it remains unclear whether a similar relationship exists when neurons are not functionally organized, but exist in heterogeneous pools. Neurons in the prefrontal cortex (PFC), for example, do not cluster anatomically by common tuning or response properties. Instead, anatomically intermingled neurons respond to multiple task variables with a plethora of temporal dynamics. Given this heterogeneity, it is not uncommon to record two neurons with entirely different response properties from the same electrode. As previous studies relating neuron spiking and high-frequency LFPs have focused on cortical areas where neural responses form organized maps, it is not clear how aggregation of PFC neurons might be reflected in high-frequency signals. As neural interface applications move beyond the motor field and into realms such as psychiatry[16], we must consider how generalizable observations from sensory and motor regions are elsewhere in the brain, and whether the cognitive information encoded by PFC neurons can be discerned from HGA.

To address this, we investigated the relationship between neuron spiking and high-frequency voltage fluctuations in a region of PFC, the orbitofrontal cortex (OFC). We focused on the OFC because of the known responsiveness of neurons in this area to motivational task features[17, 18]. The OFC is critically involved in value-based decision-making[19–21], and neuropsychiatric disorders are frequently associated with OFC dysfunction[22–24].

Using a task with a rich structure of reward-related stimuli, we determined how information was encoded by OFC neurons and high-frequency LFPs. Our results show that HGA carries abundant task-related information and reflects the aggregate activity of OFC neurons. Rather than obscuring the heterogeneous information encoded by individual neurons, however, the aggregated signal revealed spatial and temporal structure at the mesoscale that was not observed in the single neuron recordings.

## Results

**Task and behavior**. Two monkeys performed a reward expectation task that included a variety of reward-related information expected to drive neural responses in OFC. Eight familiar pictures predicted rewards of different amounts and types (Fig. 1): either primary reward (juice) or a secondary reinforcer. Pictures were probabilistically associated with trial outcomes, allowing us to distinguish encoding of expected and actual rewards. In preliminary training, amounts of juice and secondary reinforcer were titrated so that outcomes of the same ordinal value (1 to 4) were approximately equally preferred, and rewards of higher ordinal value were chosen over those of lower value regardless of reward type[25].

On each trial, subjects fixated on a reward-predicting picture for 450 ms, after which one of two response-instruction pictures appeared; one instructed the monkey to move a joystick to the right, the other to the left. These were selected independent of the preceding reward-predicting picture. If the joystick response was executed correctly, a reward was delivered. Requiring a joystick response ensured that the animals were attending to the task and confirmed that they recognized the values associated with reward-predicting pictures, as higher value pictures were followed by faster joystick responses[25].

The secondary reinforcer was a blue reward bar visible on the task screen, whose length represented the amount of juice the subject would receive after completing a trial block. The bar grew in length when monkeys received a secondary reward, and was automatically cashed in after four trials of any type. We previously found that OFC neurons encoded both secondary reinforcer amount and trial number within a block[18]. Therefore these features were also assessed when comparing neuronal and LFP encoding.

**HGA reflects local neuronal activity**. We recorded LFPs from a total of 460 electrodes (256 subject M, 204 subject N) and focused

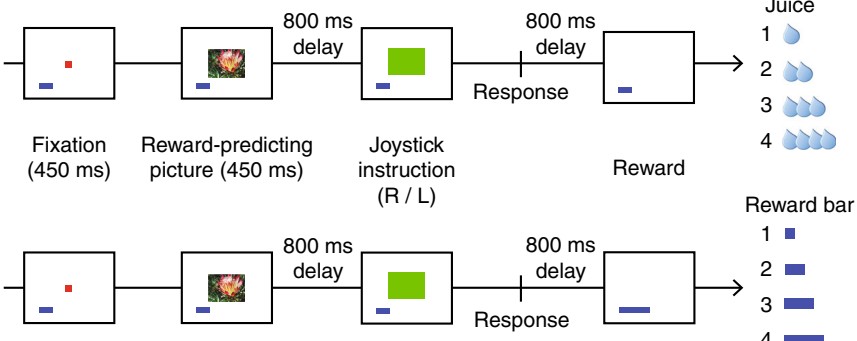

**Fig. 1** Behavioral task. To initiate a trial, subjects fixated a point in the center of the task screen for 450 ms. A picture that predicted the amount and type of reward appeared, and the subject was required to acquire and hold fixation on the picture for an additional 450 ms. Then one of two images instructed the subject to move a joystick to the right (R) or left (L). If the joystick response was executed correctly, the subject received a reward probabilistically predicted by the initial picture. Half of the pictures predicted juice reward and half predicted secondary reinforcement

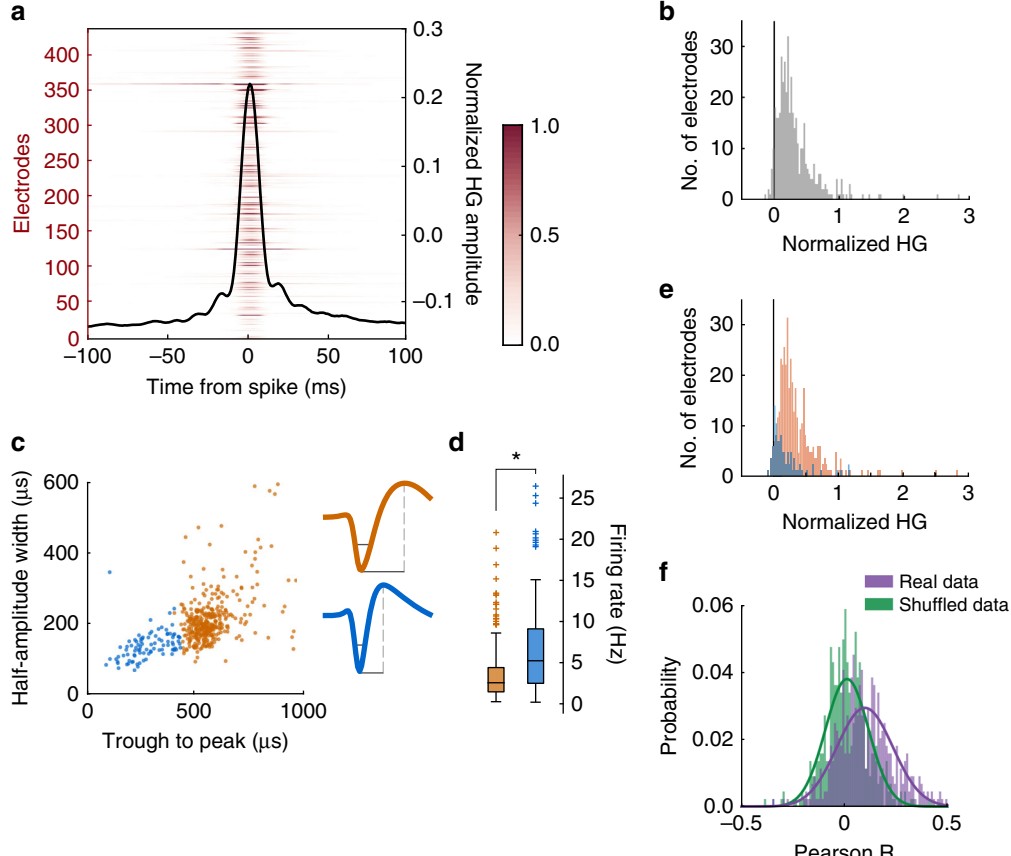

**Fig. 2** Effects of spikes on local HGA. **a** Spike-triggered HG responses for all neuron–HGA pairs recorded on the same electrode. The heatplot shows the average HGA per electrode, and the overlaid line is the average across all electrodes. HGA was z-scored to the entire recording session to obtain normalized amplitudes shown by the right y axis and colorbar. **b** Histogram of the magnitude of HGA changes observed on each electrode at the time of a spike. Changes were quantified as the number of standard deviations from the mean HGA in a pre-spike baseline epoch. Averages and standard deviations were calculated across spike occurrences. **c** All neurons were clustered into two groups (red, blue) based on their average waveforms. The mean waveform of each group is shown, and gray lines indicate the parameters used for clustering. **d** Firing rates were higher among putative interneurons (blue) than putative pyramidal cells (red). Boxes show the median, 25th, and 75th percentile of each cluster, with + indicating outlier neurons. *p < 0.001. **e** Histogram of the magnitudes of spike-triggered HGA changes, separated into groups corresponding to the clusters identified in (**c**). **f** Histogram of trial-wise partial correlations between neuron firing rate and HG amplitude when both were recorded on the same electrode (purple), or when they were recorded simultaneously on different electrodes (shuffled, green). Electrode counts were converted to proportions, and Gaussian probability functions were overlaid

on the amplitude of high-frequency responses. We chose a frequency range from 75 to 150 Hz, which optimized the detection of task-evoked responses in our set-up (Supplementary Fig. 1). These frequencies consist of a broadband signal[14] commonly referred to as "high-gamma" (HG).

We first validated that HGA in this range correlated with local neuron spiking, as expected from previous studies[8–13]. We examined neuron–HG pairs recorded from the same electrode to determine whether spiking correlated with changes in HG amplitudes ($n = 441$, 259 pairs subject M, 182 subject N, respectively). We aligned HGA to the occurrence of a spike and computed spike-triggered HG amplitudes. There was a time-locked increase in HGA when a spike occurred (Fig. 2a) that we quantified by measuring amplitudes as the number of standard deviations above a baseline measure (200 to 100 ms prior to the spike). This gave a positively skewed distribution, with most electrodes showing an increase relative to baseline (Fig. 2b). The shift was modest in magnitude with a mean and median change of 0.32 and 0.23 standard deviations, respectively. For each electrode, the mean HG amplitude ±10 ms from spike occurrence was compared across trials to the same baseline period with a t-test, and 90% of electrodes (397/441) showed a significant

increase ($p \leq 0.01$). Only nine electrodes (2%) recorded a significant decrease in HG amplitude at the time of a spike.

We also assessed whether neurons differentially contributed to HGA. Single units were separated into two clusters based on waveform characteristics suggested to differentiate pyramidal neurons and interneurons. Pyramidal neurons tend to have wider waveforms and longer trough to peak durations, whereas interneurons have narrower waveforms and shorter trough-peak times[26] (Fig. 2c). To ensure that cluster assignments were not the result of variability in unit isolation, we calculated signal-to-noise ratios (SNRs) for each neuron[27], and found no differences between the two clusters (mean SNRs for putative pyramidal cells 4.75, putative interneurons 4.65, Wilcoxon rank-sum, $n = 356$, 90 $p = 0.58$). Units in the second cluster had higher average firing rates, consistent with their designation as putative interneurons (Wilcoxon rank-sum $n = 356$, 90 $p = 1.28 \times 10^{-7}$) (Fig. 2d).

We computed the spike-triggered HGA for neurons in each cluster. Putative pyramidal cells had stronger effects, evoking a median HGA increase of 0.26 standard deviations during a spike, compared to putative interneurons, which evoked a median increase of 0.11 standard deviations (Wilcoxon rank-sum test $n = 349$, 87 neuron–HG pairs per cluster, $p = 5.9 \times 10^{-11}$)

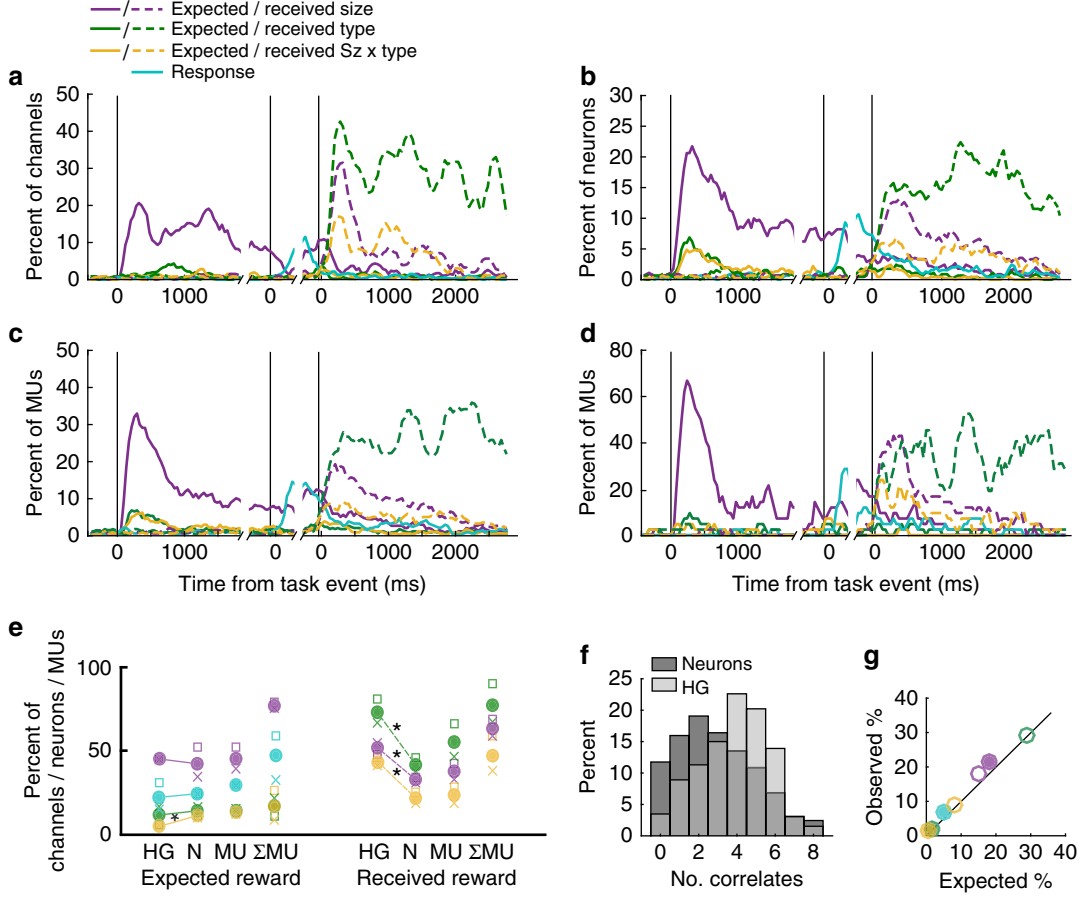

**Fig. 3** Task variable encoding in OFC HGA and single neurons. **a** Percent of channels encoding seven task-related variables as changes in HGA (significance = $p \leq 0.01 \times 3$ consecutive time bins for beta coefficients from the multiple regression). Line breaks show times where epochs were concatenated, and zero times are onset of a reward-predicting picture (first), onset of a response-instruction picture (middle), and onset of reward delivery (last). Dashed lines = variables related to received rewards. **b**–**d**. Percent of neurons (**b**), multi-units recorded from each electrode (**c**), or ΣMU (**d**) that encoded seven task-related variables, as in (**a**). **e** Percent of HG channels (HG), neurons (N), multi-units (MU), and summed MUs (ΣMU) encoding seven variables from the multiple regression in defined epochs. Color-coding is consistent with (**a**–**d**). Proportions were similar across subjects (x = subject M, square = subject N), therefore data were pooled for analysis. Pairwise $\chi^2$ tests compared neurons and HGA, and MUA is shown for comparison. *$p < 0.001$, Bonferroni-corrected for nine comparisons. **f** Number of task correlates found by multiple regression for each HG channel or neuron, across all epochs shown in (**e**). **g** Percent of HG and neurons recorded on the same electrode that also encoded the same task variable. Expected percentages were calculated from the percentage of HGA or neurons encoding each variable independently. Observed are the percentages of HG–neuron pairs with coincident encoding. Line = unity. Solid circles = variables related to expected rewards. Open circles = variables related to received rewards. Color-coding is consistent with (**a**–**e**)

(Fig. 2e). Even with these differences, the majority of single neurons of both types exhibited significant increases in HGA at the time of a spike: 96% of electrodes recording a putative pyramidal neuron showed a significant increase in HGA relative to baseline ($p \leq 0.01$ by $t$-test), as did 77% of electrodes recording putative interneurons.

Finally, if spiking contributes to HGA, trial-wise variability in firing rates should also be reflected in HGA. We assessed trial-by-trial correlations between neuron–HG pairs from the same electrode with partial correlations, where the effects of all trial variables from the multiple regression were partialed out. This approach is similar to assessing the noise correlations between neurons and HGA, which are high elsewhere in sensory cortex[13]. The outcome was compared to correlations between neurons and HGA recorded simultaneously, but where the electrode assignments were shuffled. This was done to account for any correlated activity within OFC as a whole. There were small but consistent

correlations between neurons and HGA when they were recorded on the same electrode. Effects were similar across the trial epochs, so we used the median Pearson $R$-value to compare same electrode to shuffled-electrode pairs (Fig. 2f). The distributions of $R$-values differed significantly (median 0.09 for paired, 0.007 for shuffled data, $t$-test $t_{880df} = 10.6$, $p = 6.4 \times 10^{-25}$). Therefore, there were small but consistent trial-wise correlations between HGA and firing rate that were independent of task events. Altogether, all of these approaches support the notion that HGA is related to the local neuron activity.

**Similar task encoding in HGA and single neurons.** To determine what information HGA encodes, we concatenated three task epochs and performed a sliding multiple regression analysis in windows of 200 ms, stepped forward by 40 ms. Predictors included the following: the expected size and type of reward, the

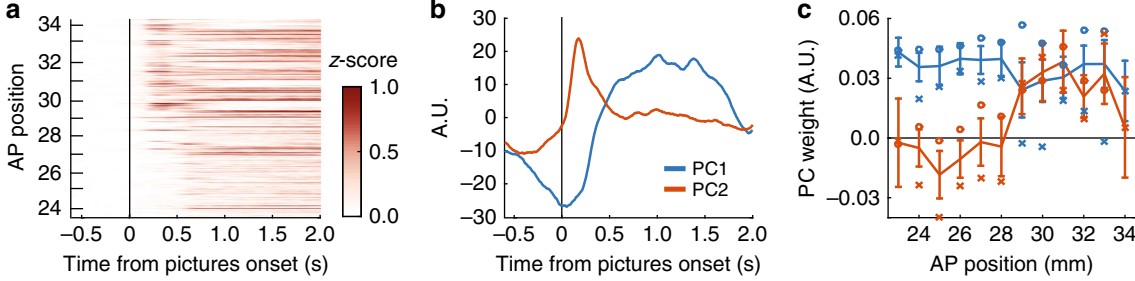

**Fig. 4** Anatomical variability in evoked HGA. **a** Average HGA responses on each electrode were sorted according to AP position, which is measured relative to the inter-aural line. Higher AP positions are more anterior. **b** The first two PCs of the data shown in (**a**). **c** Mean PC weights at each AP position. blue = PC1, red = PC2, error bars = ±95% CI. Average weights for each subject are also shown (o = subject M, x = subject N). Both subjects showed similar effects (for subject M/N: correlation of AP position with PC1 weights, $R = 0.08/-0.22$, $p = 0.18/0.002$. Correlation with PC2 weights, $R = 0.30/0.38$, $p = 8.2 \times 10^{-7}/p = 1.7 \times 10^{-8}$)

interaction expected size × type, the joystick response direction, the actual size, actual type and interaction actual size × type of reward received, the size of the reward bar prior to reward delivery, and the trial number within a block, which predicted when the bar would be cashed in for juice. Significant interaction terms indicated differential encoding of reward size, depending on the type of reward.

HGA dynamically encoded relevant information at each stage of the task (Fig. 3a). After the appearance of a reward-predicting picture, HGA encoded the expected reward size, followed by the joystick response once the instructional cue was presented, then the reward that was actually received after reward onset. The two variables that were constant within a trial but varied across trials were also encoded (Supplementary Fig. 2).

In the population of single neurons, the overall encoding pattern was similar to that of HGA (Fig. 3b). The same multiple regression analysis was conducted on each of 451 single neurons (259 subject M, 192 subject N), and revealed strong encoding of expected reward size following a reward-predicting picture (42% of neurons), then a smaller number of neurons encoding response direction (24%), then a surge in received reward size (33%) and type (41%) encoding following reward delivery.

To quantitatively compare HGA and neurons, we found the cumulative number of neurons or channels encoding each variable at any point in the relevant trial epoch. Although the patterns appeared similar, HGA had more pervasive encoding of received reward variables (Fig. 3e) ($\chi^2_{1df} > 34$ by pairwise $\chi^2$ tests, $p < 4.4 \times 10^{-8}$ after Bonferroni correction for nine comparisons). This stronger signal may be the result of HGA aggregating across many neurons, reducing the temporal and/or trial-wise variability found in single neuron firing rates, and resulting in more instances of significant encoding. If this were true, other aggregate signals such as multi-unit activity (MUA) may resemble HGA more closely than single neurons. To test this, we separated MUA, including all spikes previously categorized as single neurons, from non-neuronal noise, to arrive at a typical MUA signal ($n = 400$ MUs, 215 subject M, 185 subject N). In addition, to test another mass signal, we aggregated all spiking recorded simultaneously to create a summed multi-unit signal for each session (ΣMUA). The same regression was run with these MU signals as described for neurons and HGA. Indeed, MUA revealed patterns that resembled HGA and neurons, with encoding levels that were intermediate to the two. For example, among HGA channels, MUA channels, and neurons, 52, 37, and 33% (respectively) encoded received reward size, and 73, 54, and 41% encoded received reward type (Fig. 3c). Interestingly, ΣMUA had more encoding of all task variables than HGA (Fig. 3d). This is likely because ΣMUA combined many channels across the entirety of OFC, aggregating even more than HGA. Overall, this

supports the notion that aggregation of spiking activity increases the incidence of significant encoding in this analysis.

If HGA aggregates across neurons, we would expect more task correlates per HG channel, as if multiple neurons were added together. Since encoding was defined across epochs, a neuron or HG channel that encoded, for example, expected and received reward size, would have two correlates, as would a neuron or channel that encoded both expected reward size and trial number. Indeed, there were significantly more correlates per HG channel than per neuron (Wilcoxon rank-sum test $n = 460$, 451, $p = 1.9 \times 10^{-13}$). Neurons tended to have 1 to 3 correlates, whereas HG channels had 4 to 5 (Fig. 3f). This supports the idea that OFC HGA reflects an aggregation of multiple heterogeneous signals, such that task features linearly add to give rise to the overall HG amplitude.

Finally, we tested whether neurons and HGA recorded from the same electrode had similar encoding properties. To assess this, we found the proportion of electrodes for which both signals encoded a given variable (e.g., expected reward size), then compared this to the proportion of electrodes expected to have the same encoding in both signals by chance. For example, 42.1% of neurons and 44.8% of HGA channels encoded expected reward size. If the signals were independent, by chance 18.9% (42.1% × 44.8%) of electrodes would record neuron–HG pairs that both encode expected reward size. We compared these proportions with binomial tests (Bonferroni-corrected for nine comparisons), and found that coincident encoding was never observed more than expected by chance ($p > 0.05$) (Fig. 3g). This result seems surprising, as it appears to contradict the notion that HGA is aggregate neuronal activity. However, if a large number of heterogeneous neurons contribute to HGA, there would be a low probability that one recorded neuron drives the overall HGA encoding.

**Anatomical distribution of task encoding.** So far, our results indicate that HGA aggregates neurons with heterogeneous properties. With this in mind, we assessed the anatomical distribution of encoding to determine whether responses are, indeed, heterogeneous or if they tended to cluster. Recording sites spanned ~11 mm in the anterior–posterior (AP) dimension and were located between the medial and lateral orbital sulci, covering ~6 mm width. This includes OFC areas 11 and 13. Although we cannot identify cytoarchitectonic boundaries from structural brain scans, more anterior electrode sites should be in area 11, whereas posterior sites should be in 13. These areas differ in cytoarchitecture, with more laminar differentiation and higher cell densities in area 11[28, 29]. To determine whether differences in cytoarchitecture might change HGA generally, we first assessed picture-evoked responses at each AP electrode position. There

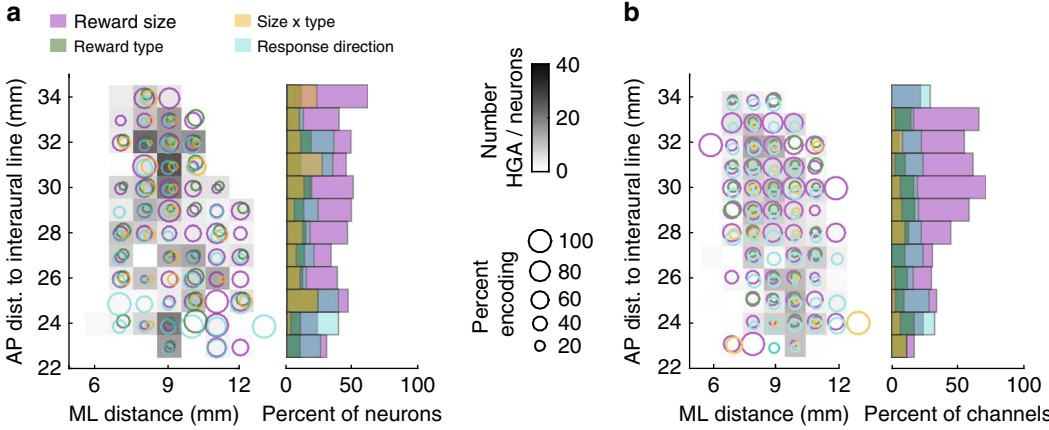

**Fig. 5** Anatomical distribution of task encoding. **a** A map of recording sites in OFC. The gray shading of each square represents the total number of neurons recorded at each position in the AP and ML dimensions. Colored circles indicate encoding of pre-reward task variables, and diameters are proportional to the percent of neurons at each location that encoded the indicated variable. Bar plots in the second panel show the aggregate data for each AP position. **b** The same map of OFC recording sites as (**a**), except for HGA

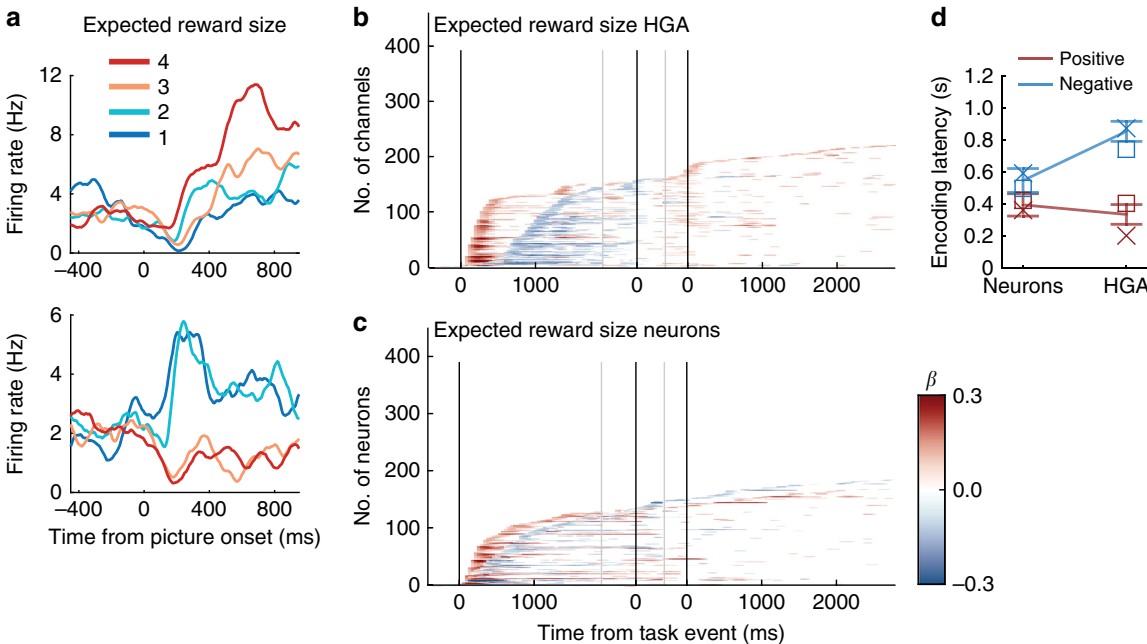

**Fig. 6** Valence of expected reward size encoding. **a** Firing rates of two example neurons recorded from the same electrode. Both encode expected reward size, but with opposing valence. **b** Significant encoding of expected reward size over time in HGA. The x axis shows time concatenated across the three epochs, picture onset (first), response (middle), and reward receipt (last), as in Fig. 3. Electrodes with positive (red) and negative (blue) encoding valence were sorted separately and overlaid. **c** The same plot as (**b**), but for single neurons. **d** Mean latencies to encode expected reward size ± 95% CI, separated based on whether the response was higher for larger (positive, red) or smaller rewards (negative, blue). Mean latencies for each subject are also shown: x = subject M, square = subject N

were two clear patterns, an early phasic response peaking ~250 ms after picture onset, followed by a more sustained response beginning at ~500 ms. These two components were also apparent in the averaged signal in Supplementary Fig. 1b.

When sorted by AP position, the early phasic response was only present in anterior electrodes, but the sustained response was present throughout OFC (Fig. 4a). To quantify this, principal component (PC) analysis extracted the first and second PCs across electrodes, which correspond to the two response profiles (Fig. 4b), and together accounted for 63% of response variance. PC weights measured how strongly each response profile was manifest on a given electrode. For example, electrodes with an early phasic component had higher PC2

weights. Indeed, PC2 weights (early phasic responses) increased at more anterior positions, whereas PC1 weights (sustained responses) did not depend on position (Fig. 4c). Pearson correlation with AP position was significant for PC2 ($n = 460$, $R = 0.33$, $p = 3.6 \times 10^{-13}$), but not PC1 ($R = -0.07$, $p = 0.12$). These results suggest that HGA varies across electrodes and may be a useful measure for identifying spatiotemporal signatures across different regions of cortex. The early phasic responses also coincided with a negative potential at ~240 ms in the broadband LFP that was larger among anteriorly placed electrodes (Supplementary Fig. 3).

To assess encoding of task variables at different anatomical locations, electrode locations were collapsed onto an AP by

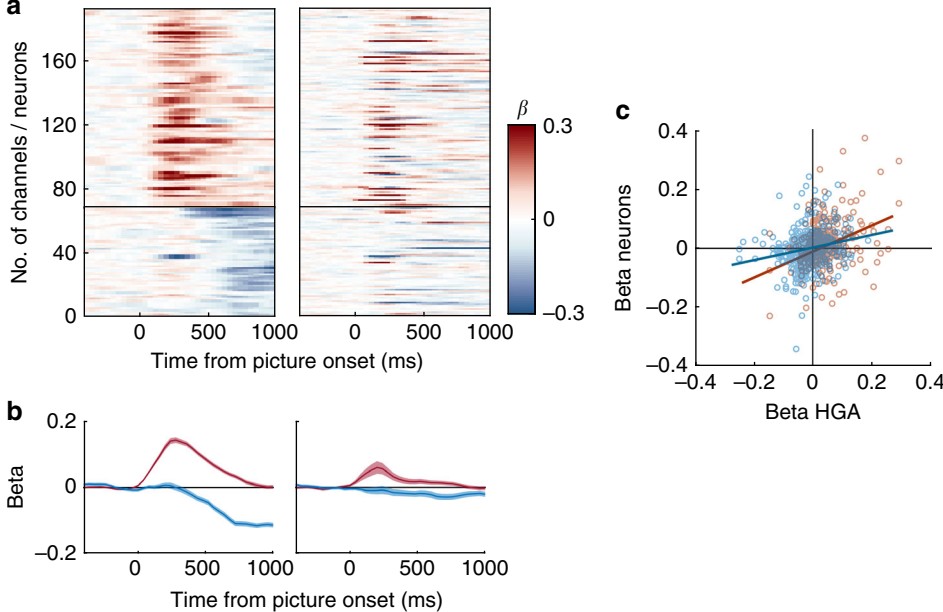

**Fig. 7** Valence responses in signals on the same electrode. **a** The left panel shows beta coefficients for all HG channels that significantly encoded expected reward size. Those with positive encoding (red) are above the line and negative encoding (blue) are below. The right panel shows beta coefficients for neurons recorded from the same channels as the HGA in the left panel. **b** Average beta coefficients from (**a**), with signals grouped based on whether the HGA on that electrode encoded reward size positively (red) or negatively (blue). **c** Average beta coefficients for all recorded HG–neuron pairs in the first 500 ms after picture onset (red), and from 500–1000 ms after picture onset (blue)

medial–lateral (ML) grid by combining across hemispheres, based on the AP position and distance lateral to the midline. We first assessed the percent of neurons at each location encoding task variables prior to feedback, including the expected size and type of reward and the direction of the joystick response. Neurons and HGA encoding pre-reward variables were found throughout OFC (Fig. 5). To quantify encoding across AP locations, we collapsed across the ML dimension to find the total proportion of neurons or HG channels encoding each variable at sites in anterior OFC (AP 28 to 34, relative to the inter-aural line) and posterior OFC (AP 23 to 27) with $\chi^2$ tests (Bonferroni-corrected for nine variables tested). Among neurons, there was a weak tendency to encode expected reward size anteriorly ($\chi^2_{1df} = 8.31$, corrected $p = 0.04$), but this did not reach significance in either subject individually (both $\chi^2_{1df} < 4.5$, corrected $p > 0.3$). However, in HGA, encoding of expected reward size was much stronger anteriorly ($\chi^2_{1df} = 37.7$, corrected $p = 7.4 \times 10^{-9}$), and the effect was significant in both subjects individually (both $\chi^2_{1df} > 14.4$, corrected $p < 0.002$). No other pre-reward variables differed between anterior and posterior OFC among neuron or HG populations ($\chi^2_{1df} < 3$, corrected $p > 0.9$). Therefore, at a large anatomical scale, HGA encoding patterns were more pronounced than those observed among single neurons. We performed the same analyses on task variables related to reward receipt, as well as the reward bar length and trial number within a block, and found similar though less marked effects, whereby HGA encoding showed stronger anatomical trends than single neurons (Supplementary Fig. 4).

**HGA and neurons differ in encoding valence**. It is well documented that roughly half of OFC neurons that encode stimulus value do so directly, by increasing firing rates for higher value items, whereas the other half do so inversely, by increasing firing rates for lower value items[17, 30–32]. Such opposing schemes are common to many features encoded in PFC[33–35]. An example

from the current dataset is in Fig. 6a, which shows two units recorded simultaneously from the same electrode. If the activity of neurons with opposite encoding schemes were aggregated, one might expect weaker or no encoding in HGA, rather than stronger encoding as we observed here.

To investigate how the valence of encoding in HGA relates to single neurons, we looked at the beta coefficients from the multiple regressions above. Figure 6b shows significant betas for expected reward size for all HG channels, sorted by encoding latency. Positive betas, which indicated higher amplitudes for larger reward sizes, were sorted separately from negative betas, which indicated higher amplitudes for smaller reward sizes. There were two temporally distinct volleys of expected reward encoding with positive encoding occurring first, followed ~500 ms later by negative encoding. Separate channels encoded reward size in different directions, and few channels (~5%) flipped from positive to negative later in the trial.

Similar positive and negative responses were evident in the single neurons, but with different temporal dynamics (Fig. 6c). To compare encoding latencies between the two signals, we conducted a 2×2 ANOVA that revealed an interaction between the signal (HGA vs. neurons) and encoding valence (positive vs. negative) ($F_{1,522} = 29.2$, $p = 1 \times 10^{-7}$) (Fig. 6d). There was no latency difference between HGA and neurons with positive encoding (post hoc analyses of simple effects within valence $F_{1,522} = 1.6$, $p = 0.2$), but a large difference between HGA and neurons with negative encoding (post hoc analyses $F_{1,522} = 38.7$, $p = 1 \times 10^{-9}$), suggesting a disparity between the two signals. In this case, it appeared as if there was no neuronal correlate of the negative HGA encoding volley.

To further understand this relationship, we looked at the overall variability in neuron firing rates, regardless of whether they reached significance in our encoding tests. We identified HG channels with significant encoding of expected reward size, as well as the neuron(s) recorded from the same electrode, and assessed the similarity in their beta coefficients. HG channels

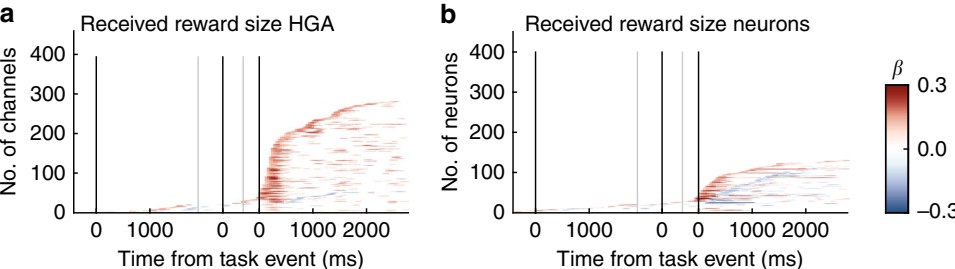

**Fig. 8** Valence of received reward encoding. **a** Significant encoding of received reward size over time in HG amplitudes as in Fig. 6b. **b** The same plot as (**a**), but for single neurons. red = positive, blue = negative encoding valence

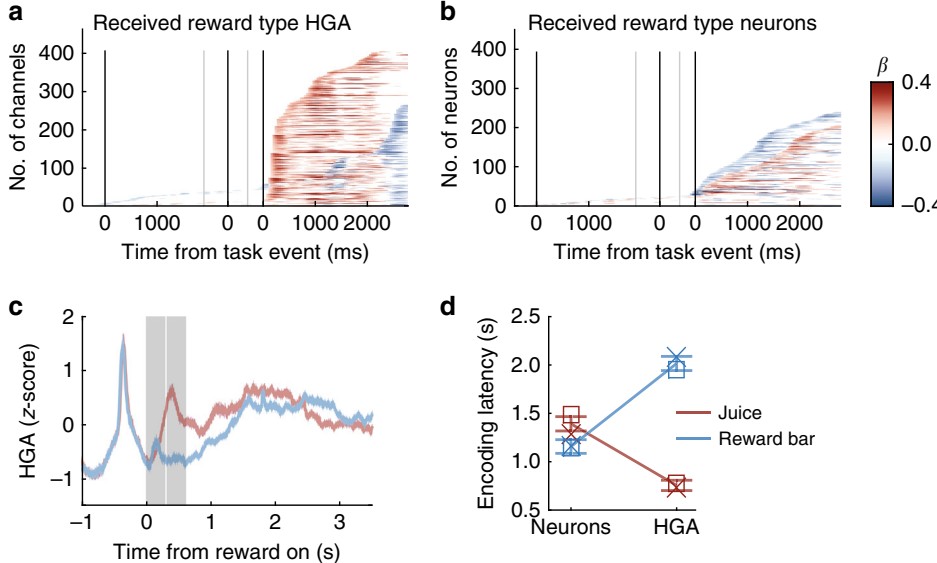

**Fig. 9** Received reward type encoding. **a** Significant encoding of received reward type over time in HGA and **b** single neurons as in Fig. 6b. Red = juice-preferring, blue = secondary-preferring. **c** Mean ± SEM HGA on all trials where the subject received juice (red) or an increase in reward bar length (blue) as a reward. Gray shaded regions show the time when the juice pump was running (2 × 300 ms epochs). The HG response to juice reward does not align temporally to the two juice boluses, so the difference in response is not related to electrical noise from the pump. **d** Mean latencies to encode reward type ± 95% CI, separated based on whether the signal was higher for juice or secondary reward. Mean latencies for each subject: x = subject M, square = subject N

either encoded reward size positively in the first 500 ms or negatively about 500 ms later (Fig. 7a). Neurons had more heterogeneous responses, but tended to have the same valence as HGA recorded from the same channel. If the signals were unrelated, we would expect the average neuronal response would not vary depending on HGA. However, Fig. 7b shows that, even though there were not exclusive mappings, the average beta coefficients follow similar patterns.

To quantify the relationship between encoding valences, we correlated beta coefficients between all neuron–HG pairs in two 500 ms epochs (0–500 ms and 500–1000 ms from picture onset). In both cases, there was a small but positive relationship ($n = 441$ pairs, $R = 0.27$, $0.17$ and $p = 9.1 \times 10^{-9}$, $8.4 \times 10^{-5}$, respectively), even though the coefficient values shifted from more positive betas in the first 500 ms to more negative betas in the second (Fig. 7c). Therefore, when we considered all variance in neuron firing rates, including that which did not reach statistical significance, relationships to HGA were revealed, consistent with the notion that HGA reflects an aggregated and amplified signal from single neurons. Indeed, MUA also showed distinct temporal patterns for positive and negative encoding (Supplementary Fig. 5a–c). Thus, aggregation in HGA or MUA revealed more temporal structure in OFC value encoding than we could obtain

from single neuron activity. Comparing MUA and HGA recorded from the same electrode yielded a pattern intermediate to single neurons and HGA, again as would be expected if MUs aggregate over smaller populations of neurons than HGA (Supplementary Fig. 6a–c).

**Received reward size encoding**. To determine how generalizable these results are, we analyzed neuron and HGA responses during reward receipt. Neurons encoded received reward size both positively and negatively in approximately equal proportions (cumulatively 24% positive, 21% negative). In contrast, a large proportion of HG channels encoded reward positively (59%) and very few showed negative encoding (8%) (Fig. 8). More homogeneously positive encoding in HGA could be a result of aggregating across single neurons, where positive encoding was slightly more prevalent. Indeed, MUA on single electrodes showed an intermediate pattern, with a bias toward positive encoding, whereas ΣMUA had nearly entirely positive encoding (Supplementary Fig. 5d, e), suggesting that the positive bias increased with increasingly aggregate signals. Because HGA lacked appreciable negative encoding, we did not analyze these responses further.

With regard to encoding the type of reward received, we found pronounced differences between neurons and HGA (Fig. 9a, b).

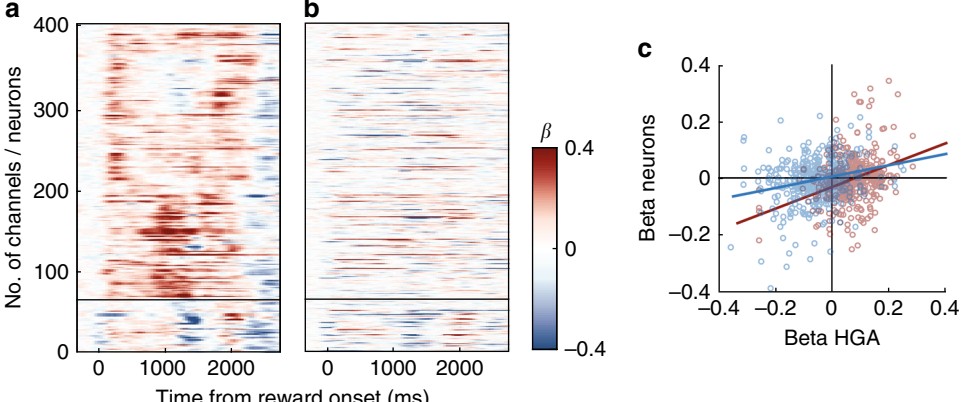

**Fig. 10** Reward type encoding on the same electrode. **a** Beta coefficients for all HG channels that significantly encoded received reward type. Those with stronger average responses to juice (red) are above the line and reward bar (blue) are below. **b** Beta coefficients for neurons recorded from the same channels as the HGA in the left panel. **c** Average beta coefficients for received reward type in all recorded HG–neuron pairs in the first 500 ms after reward onset (red), and from 2500–3000 ms after picture onset (blue)

Here the sign of beta coefficients denoted whether responses were stronger for primary or secondary rewards. Approximately equal proportions of neurons responded preferentially to each reward type (42% juice, 50% reward bar), but a majority of HG channels initially preferred juice rewards (85%), and later favored secondary rewards (52%). This shift occurred ~2 s after reward onset, during the inter-trial interval. The average time course of HGA showed that responses to juice reward began earlier and decreased earlier than responses to secondary rewards, resulting in beta coefficients that changed over time. Mean encoding latencies were 755 and 2016 ms when HGA was higher for juice and reward bar, respectively (Fig. 9c). A 2×2 ANOVA of latencies revealed an interaction between signal (HGA vs. neurons) and reward type ($F_{1,1046} = 265.5$, $p = 2.3 \times 10^{-53}$). Simple effects analyses found that all contrasts were significant (all $F_{1,1044} > 11$, $p < 9 \times 10^{-4}$), with neurons responding earlier to secondary rewards and HGA responding earlier to juice reward, the latter of which produced the most significant contrast (Fig. 9d). The different responses to juice and reward bar may be an effect of the different sensory modalities carrying reward information, or additional processing required for interpreting secondary rewards. In this case, MUA did not approximate HGA patterns (Supplementary Fig. 5f–h).

As in the earlier epoch, we assessed whether variability in neuron firing rate, regardless of regression significance, correlated with HGA on the same electrode. Beta coefficients for HG channels with significant encoding of reward type within the 3 s after reward, as well as the neuron(s) recorded on the same electrode ($n = 407$ pairs), are shown in Fig. 10. On most HG channels there was a rapid response to the receipt of juice reward that was sustained for variable durations, followed ~2 s later by higher activity on trials when the subject received a secondary reward (Fig. 10a). A similar pattern was not clearly discernable among single neurons (Fig. 10b), nor MUA (Supplementary Fig. 6d). Despite the heterogeneity of spiking data, there were consistencies with HGA. Pearson correlations between average beta coefficients for reward type in pairs of HGA and neurons recorded from the same electrode were weak but positive in both the first 500 ms after reward onset, when most HG responses were stronger on juice reward trials ($n = 407$ pairs, $R = 0.24$, $p = 3.1 \times 10^{-7}$) and in 500 ms starting 2.5 s after reward onset, when most HG responses were stronger on secondary reward trials ($R = 0.32$, $p = 2.1 \times 10^{-11}$) (Fig. 10c). MU activity submitted to the same analysis showed similar correlations with HGA as single neurons (Supplementary Fig. 6e). Overall, the responses to

different types of rewards provided the biggest distinction between spiking activity and HGA, with more response heterogeneity in spiking data, though there were some consistencies between signals.

## Discussion

The main question we aimed to address was whether PFC neurons and HGA encode similar information during cognitive processing. We focused on the OFC and used a task with a rich structure of motivational cues that would produce diverse encoding in these neurons. Taken together, our results revealed a high degree of similarity in the information encoded by HGA and single neurons, and the dominant variables encoded by both signals reflected information pertinent to the task at each point in time. Previous work has shown that HGA is a reliable signal for mapping regional activation in the brain[2], and a good deal of evidence suggests that HG responses reflect the aggregated spiking of local neurons[8–13]. However, these investigations have primarily been carried out in sensorimotor cortical regions where neurons exhibit some degree of functional clustering, which appears to be a critical factor in LFP encoding[8]. It was unknown whether heterogeneity in PFC would result in HGA that is less selective for task information than corresponding neuron populations. Our results show that this is not the case. Encoding in HGA was slightly stronger than among neuron populations, suggesting a more widespread or less noisy signal, consistent with earlier reports of information fidelity on par with or better than that observed in single neurons[6, 36]. Overall, these results show that it is reasonable to use HGA to infer dynamics of information encoding across brain regions[2] including PFC, underscoring HGA as an important signal for understanding and interfacing with brain functions[7].

Although there was a good deal of similarity between neurons and HGA encoding at the population level, this did not extend to hyperlocal pairings of neurons and HGA recorded from the same electrode. The variable(s) encoded by a given neuron did not predict those encoded by the corresponding HGA. In some respects, this seems counter-intuitive since we also claim that local neurons contribute to HGA. To make sense of this finding, we must consider the functional heterogeneity of neurons in OFC. Regions of PFC are broadly distinguished by trends in physiological responses[18, 37, 38], but these differences cover large areas and have not supported the notion that there are sub-regional maps analogous to those in sensory or motor cortex.

Instead, information encoded in PFC tends to be intermingled, with multiplexing[39] or mixed selectivity[40, 41] described at the single neuron level. These features may contribute to PFC's computational abilities. Some have argued that, because of this signal mixing, task-related responses in PFC provide an inferior metric for anatomical parcellation, compared to trial-wise correlations in neuron activity[42]. OFC neurons encode a variety of task features with a mix of encoding valence[18, 32, 37, 43–45] and task-evoked responses have revealed little if any anatomical organization within OFC areas 11 and 13[18, 32, 44, 45].

Although these properties of OFC neurons are well established, the results presented here are the first thorough investigation of HGA in non-human primate OFC. Elsewhere in the brain, it is estimated that the LFP recorded from a microelectrode reflects neuronal activity within ~250 μm of the tip[46], though this may increase with coordinated neural activity[47]. Given average cell densities in PFC[29], this means that at least 2000 to 3000 neurons contribute to HGA on one electrode. Although neuron contributions are thought to scale with distance to the recording electrode[1], in a pool of this size the contribution of any given neuron to the overall HG response would remain small. In agreement with this, we found consistent contributions of single neuron spiking to HGA recorded on the same channel. But these contributions were small, so that a single neuron did not drive overall encoding properties of HGA.

In contrast to our results, HGA correlates with population responses when neurons with similar tuning properties are anatomically clustered[8, 9, 48], and there is evidence that this correlation is based on the spatial scale of functional groupings[8]. It is likely that signals existing over larger spatial extents have more impact on high frequencies, and contribute to a broader range of frequencies. This may explain why encoding of expected reward size, for example, is found among many OFC neurons and appears as a robust signal in HGA, whereas expected reward type is only encoded by a small and distributed number of neurons and not found in HGA. This encoding may be obscured by the more prevalent and widespread signals.

A prominent feature of our data was that HGA revealed clearer spatial and temporal structure than single neurons. Anatomically, the biggest distinctions were between anterior and posterior OFC. Few studies have reported AP differences in neuronal responses[49], and in agreement we find very weak differences in our population of neurons. However, there were pronounced differences in HGA encoding, with some task correlates, particularly expected reward size, more prevalent in anterior OFC. The anterior region also exhibited early phasic responses that were not found in posterior OFC sites. This regional variation may correspond to anterior area 11 and posterior area 13, which are distinguished by differences in cytoarchitecture[29, 50] and connectivity[51, 52], and can be functionally dissociated in neuropsychology studies[53, 54]. Murray et al.[53] recently reported a double-dissociation in macaque OFC, with areas 13 and 11 uniquely required to update and use stimulus-value information, respectively. These results are consistent with our finding that learned value information is preferentially represented in anterior OFC. Importantly, our posterior OFC sites, while lacking the early phasic responses found in anterior OFC, still exhibited robust HG sustained responses on par with anterior sites, and the different cytoarchitecture in posterior OFC did not result in an overall degraded HGA signal.

In addition, HGA revealed temporal structure that was not apparent among single neurons. This was illustrated with respect to encoding valence, that is, whether responses were stronger for more or less valuable pictures and rewards. There were roughly equal proportions of neurons that were more active for large rewards and for small rewards, consistent with previously described data[18, 32, 37, 43–45]. These two populations were mostly overlapping in time course, with slightly earlier responses among neurons with positive valence. In contrast, HGA had more structure. When subjects viewed a reward-predicting picture, there were two volleys of HG encoding. The first included channels that responded more to large rewards followed ~500 ms later by channels responding more to small rewards. The two volleys overlapped with the early phasic and late sustained patterns found in evoked responses. The significance of these two volleys of activity remains to be determined, but may reflect the activity of different long-range circuits interacting with OFC[55].

At the time of reward receipt, HGA and neuronal signals exhibited further differences. There were almost no HG channels that responded more for small rewards, despite the fact that nearly half of single neurons did. This was in contrast to the bivalent responses elicited by reward-predicting pictures. Further, slightly more single neurons preferentially responded to secondary rewards, whereas nearly all HG channels initially responded more to primary rewards, followed by a delayed response to secondary rewards.

How, then, can we reconcile the different patterns that emerged from these two signals with the notion that HGA reflects neural spiking[8, 9, 48]? One explanation might lie in the relationship we observed between HGA and neuronal encoding when the regression models were not thresholded for significance. In this case, statistically significant trends observed in HGA were weakly apparent in neurons recorded from the same electrode. That is, neurons that responded to a reward-predicting picture with non-significant increases (or decreases) in firing rate tended to be recorded from electrodes where HGA encoding was also positive (or negative). Since one recorded neuron represents only a small proportion of the population contributing to HGA, consistent valence relationships between one neuron and HGA suggest that these trends are a pervasive feature of the local neuron population. Even small changes in firing rate could create robust HGA encoding when present across a population of neurons. One possible mechanism for these small but pervasive shifts is subthreshold voltage fluctuations that affect collections of neurons, making each more or less likely to emit a spike at a given time[56]. Although this would create only small changes in a given neuron, the aggregate effect in a population would be larger. Indeed, the LFP is known to correlate with membrane potentials of neurons in the vicinity of the electrode[56], and synchronous activity is believed to make strong contributions to HGA[1, 11], potentially explaining how small changes in the spiking probability of individual neurons could result in large modulations of HGA.

Overall, our results show that high-frequency LFP signals recorded from OFC were related to the spiking activity of local neuronal populations, consistent with results elsewhere in cortex[8–13]. Potentials in the HG range provided a high quality signal that robustly encoded similar information as single neurons. In those cases where HGA appeared to diverge from neuronal activity, the disparities could be explained by smaller shifts in neural firing rates that were not considered significant in formal encoding analyses. In addition, HGA provided more insight into spatial and temporal variability at the mesoscale, making it a promising signal for understanding functional organization across cortex, and bridging single unit recording in animals and neurophysiology in humans.

## Methods

**Subjects and behavior**. Subjects and behavior have been described in a previous publication that investigated network-level mechanisms of choice[25]. Here we focused on separate trials that included only a single reward-predicting picture. All procedures were in accord with the National Institute of Health guidelines and recommendations of the University of California at Berkeley Animal Care and Use Committee. Subjects were two male rhesus macaques (*Macaca mulatta*), aged 7 and 9 years, weighing 14 and 9 kg at the time of recording. Subjects sat in a primate

chair, viewed a computer screen and manipulated a bidirectional joystick. Task presentation and reward contingencies were controlled using MonkeyLogic soft-ware[57], and eye movements were tracked with an infra-red camera (ISCAN, Woburn, MA).

Reward-predicting pictures were 8 familiar images of natural scenes, ~2° × 3° of visual angle. Pictures were selected randomly from this set on each trial. Four pictures predicted the delivery of juice reward (0.05, 0.10, 0.18, 0.30 ml), and four predicted that the length of the reward bar would increase by a set increment (Fig. 1). Prior to task training, subjects were conditioned to associate the length of the reward bar with a proportional amount of juice obtained at the end of a trial block. When given a choice, subjects M and N chose pictures that predicted larger over smaller gains on 91 and 97% of choices, respectively[25]. After four completed trials, the reward bar was automatically exchanged for the corresponding amount of juice and reset to a small initial size. This exchange occurred regardless of the trial type or outcome of the trials in the block.

Outcomes associated with the pictures were probabilistic. On 4/7 trials (~57%), the actual reward amount and type were consistent with the reward-predicting picture. On 1/7 trials (~14%), the actual reward type was consistent with the predicted reward type, but the actual reward amount was one of the three other values. On 1/7 trials (~14%), the actual reward amount was consistent with the predicted reward amount but the actual reward type was the opposite of the predicted type. Finally on 1/7 trials (~14%), both reward amount and type were inconsistent with the reward-predicting picture.

The single-picture trials analyzed here, in which only one reward-predicting picture was shown, were randomly interleaved with choice trials in which two pictures were randomly selected and the subject was allowed to choose one and receive the corresponding reward. Choice trials have been described in detail elsewhere[25]. Briefly, both subjects preferred pictures of higher ordinal value, and reward amounts were titrated so that preferences were approximately equal across different types of reward. Sessions in which < 300 trials were completed were excluded to ensure sufficient sampling of neural responses (3 subject M, 1 subject N). A total of 44 recorded sessions were included (24 subject M, 20 subject N).

**Neurophysiological recording.** Subjects were implanted with head positioners and two titanium chambers positioned to access OFC bilaterally. Up to 16 electrodes were acutely lowered to OFC through craniotomies following previously described methods[25]. Electrodes targeted areas 11 and 13 based on previously obtained MR images and acoustically mapping gray and white matter boundaries during low-ering of electrodes. Neurons were not screened for selective responses, and all well-isolated neurons in the target region were recorded and included in the analyses. If a neuron was not isolated, the electrode was left in the cortical layer to record LFP and MU activity.

Neural signals were acquired with a Plexon MAP system (Plexon, Dallas, TX), with all signals referenced to ground, which contacted skull screws. The continuous signal was split into two streams online by the preamplifier. The LFP stream was high-passed at 0.7 Hz then low-passed at 300 Hz and digitized at a rate of 1 kHz. The spike stream was high-passed at 100 Hz and low-passed at 8 kHz, and waveforms crossing threshold were digitized at 40 kHz and stored, with gains and thresholds manually adjusted on a channel-by-channel basis. High-pass filters and low-pass filters were 2-pole and 4-pole Butterworth, respectively.

**Neural signal processing.** Single units and MUs were separated offline (Offline Sorter, Plexon). Initially, single units were isolated as spikes that formed distin-guishable clusters with consistent waveform shapes and ≥ 99.8% inter-spike intervals longer than 1400 μs. Then, remaining low-amplitude waveforms were separated from noise clusters and artifacts, and combined with isolated single units on the same electrode to create MU clusters. If low-amplitude waveforms were not well separated from non-neuronal noise, these clusters were excluded. For analysis, spike times were transformed to a time series with 1 kHz resolution, where 1 indicated the presence of a spike, and 0 the absence. For encoding analyses, this time series was smoothed with a 50 ms boxcar and z-scored across the entire session.

Raw LFP signals were first evaluated visually and channels with excessive artifact, or in which the voltage was clipped by the amplifier, were removed. The remaining channels were notch filtered at 60, 120 and 180 Hz before being submitted to further analyses.

**Data analysis.** For the power spectrum analysis, we used a 4 s time window around the appearance of a picture (−1 to 3 s) on all correctly completed single-picture trials, and averaged the spectrum across trials for each electrode. Chronux software[58] was used to obtain a multitaper estimate of the power spectrum above 50 Hz. Frequencies beyond 300 Hz were removed by hardware filter and therefore not recoverable. Because the Butterworth filter produces minimal distortion below the low-pass limit, we assessed frequencies up to 275 Hz. We used five tapers and a time bandwidth product 3.

For the remaining analyses of the LFP as a time series, LFPs were band-passed using a finite impulse response filter and analytic amplitudes were obtained from the Hilbert transform of the pass bands. Artifacts were removed by thresholding the amplitude time series within ±7 standard deviations of the overall mean. For

evoked response and encoding analyses, amplitudes were smoothed with a 50 ms boxcar, z-scored, and detrended across the entire session. The resulting time series was aligned to task events. For the spike-triggered analysis, the data were not smoothed, and we used a random sample of 10,000 emitted spikes for high-firing rate neurons to reduce computational processing time.

For encoding analyses, time series of spikes or HG amplitudes were aligned to three task events to extract an epoch of interest: (1) the appearance of a reward-predicting picture (−500 to 2000 ms), (2) the appearance of the response-instruction picture (−500 to 500 ms), and (3) the delivery of reward (juice or reward bar, −300 to 3000 ms). The three epochs were concatenated to yield an overall evolution of the task with neural activity aligned to events. In the multiple regression analysis, all reward size variables were ordinal (1 to 4) and re-centered around zero, and reward type was dummy coded as −1 for secondary reward and +1 for primary reward. We defined significant encoding as three consecutive 200 ms bins with a regressor weight that was significantly non-zero at $p \le 0.01$. Encoding latencies were defined as the mid-point of the first of these significant windows, resulting in ≤ 2% of neurons or channels with encoding latencies <0 ms. Encoding during each epoch was summarized as the cumulative percent of neurons/channels encoding each variable in a specified time window following a task event. These were as follows: 1000 ms after appearance of a reward-predicting picture for expected variables (expected reward size, type and size × type interaction), 500 ms after appearance of a response-instruction picture for response direction, 1000 ms after reward onset for received reward variables (received reward size, type and size × type interaction), and the full time series of three concatenated epochs for reward bar size and trial number. Encoding patterns were similar in both subjects (Fig. 2e), and therefore data were pooled.

To cluster neurons into putative pyramidal cells and interneurons, we used the average waveform of each unit and found the time from the trough to the following peak, as well as the width of the negative segment of the waveform at half its amplitude[26]. Cells were clustered using k-means, optimized by randomly initiating the algorithm 1000 times and selecting the solution that minimized the sum of squared distances to the assigned cluster center. SNRs were calculated as

$$\text{SNR} = \frac{\max(\overline{W}) - \min(\overline{W})}{2 \times \text{SD}_\varepsilon},$$

where $\overline{W}$ is the mean waveform and $\text{SD}\varepsilon$ is the standard deviation of the matrix of noise values[27].

For trial-by-trial correlations, correlations were performed in sliding windows of 200 ms, stepped forward by 40 ms, over the same periods as the multiple regression. The data were first detrended by removing the best-fit linear trend across trials from the average measure in each time window to remove potential effects of electrode drift that could inflate the correlation.

To create anatomical maps of OFC recording sites, the distance anterior to the inter-aural line and lateral to midline was found for the region of cortex contacted by an electrode lowered from each site in the recording chambers, based on pre-surgical MR images. The two hemispheres of each animal were then combined to create one grid-based map of OFC. As these maps showed that the most variability in neural responses occurred between anterior and posterior OFC, statistical comparisons were performed by finding the total number of HG channels or neurons recorded at AP positions ≥ 28 mm (the mid-point of the recorded region) and < 28 mm, finding the proportion in each group with a given encoding property, and performing a $\chi^2$ test to determine whether the proportion differed in anterior and posterior OFC. When the same encoding properties were tested in multiple ways, the $p$-values were Bonferroni-corrected.

**Data availability.** The data that support the findings in this study will be made available from the corresponding author upon reasonable request. Code for these analyses is also available from the corresponding author upon request.

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

## Acknowledgements

This work was funded by NIDA R01 DA19028 and NIMH R01 MH097990 to J.D.W., and by the Hilda and Preston Davis Foundation and NIDA K08 DA039051 to E.L.R. This research was partially funded by the Defense Advanced Research Projects Agency (DARPA) under Cooperative Agreement Number W911NF-14-2-0043, issued by the Army Research Office contracting office in support of DARPA'S SUBNETS program. The views, opinions, and/or findings expressed are those of the author(s) and should not be interpreted as representing the official views or policies of the Department of Defense or the U.S. Government.

## Author contributions

E.L.R and J.D.W. designed the experiment. E.L.R. collected and analyzed the data and wrote the manuscript. J.D.W. edited the manuscript.

## Additional information

**Competing interests:** The authors declare no competing financial interests.

