## [Peer Review File · Nature Communications]

Reviewers' comments:

Reviewer #1 (Remarks to the Author):

I am satisfied with the changes made by the authors. The revised manuscript makes an interesting contribution to the interpretation of mass signals

Reviewer #2 (Remarks to the Author):

Rich and Wallis recorded from the orbitofrontal cortex (OFC) of monkeys who were engaged in a task requiring complex decision-making and reward estimation, and compared the performance of high-gamma (HG) activity and spiking activity in coding various aspects of the task. They show that HG responses, in many cases, were consistent with an aggregation of many heterogenous neurons, and were often more reliable predictors of the behavior. They also show that some aspects of the task were better captured by the HG activity.

I have previously reviewed this manuscript. The authors did incorporate many of my suggestions, and I feel the manuscript is much stronger now. Although perhaps a little light on the novelty factor, overall this is a very solid manuscript, contains recordings from a brain area that is not that well studied, involves a rich and complex task, and contains several new points that have not been well documented. So overall I recommend the study for publication. I have only minor comments on this version, as follows.

1. I find the analysis of summed neuronal activity, as shown in Supplementary Figure 2, very interesting, since it seems to outperform the HG in some cases. I think it should be moved to the main text. Further, the same analysis should be performed for the later sections (Figure 6 onwards).
2. A very interesting extension could be to use the summed activity of not the entire neuronal data, but instead only some electrodes around the LFP electrode, and to quantify the area of the summed neuronal activity that matches the HGA activity. I think this would be a valuable addition that would significantly improve the impact of the paper.
3. Supplementary Figure 1c – HG activity appears to decrease before the picture onset and reaches a minimum at 0. Why is the HG higher before the onset? Do you see the same trend in the firing rates?
4. Supplementary Figure 3 - can the authors also show firing rates as a function of AP as well?

Response to reviewers

We thank both reviewers for their feedback on the manuscript, and are pleased that reviewer #1 has no further concerns. Below we respond to the few remaining comments from reviewer #2.

Reviewer #2 (Remarks to the Author):

Rich and Wallis recorded from the orbitofrontal cortex (OFC) of monkeys who were engaged in a task requiring complex decision-making and reward estimation, and compared the performance of high-gamma (HG) activity and spiking activity in coding various aspects of the task. They show that HG responses, in many cases, were consistent with an aggregation of many heterogeneous neurons, and were often more reliable predictors of the behavior. They also show that some aspects of the task were better captured by the HG activity.

I have previously reviewed this manuscript. The authors did incorporate many of my suggestions, and I feel the manuscript is much stronger now. Although perhaps a little light on the novelty factor, overall this is a very solid manuscript, contains recordings from a brain area that is not that well studied, involves a rich and complex task, and contains several new points that have not been well documented. So overall I recommend the study for publication. I have only minor comments on this version, as follows.

1. I find the analysis of summed neuronal activity, as shown in Supplementary Figure 2, very interesting, since it seems to outperform the HG in some cases. I think it should be moved to the main text. Further, the same analysis should be performed for the later sections (Figure 6 onwards).

We have moved these analyses from Supplementary Figure 2 to the main text as requested. However, we felt that adding these data made the figure crowded, and made two changes to improve aesthetics and readability. First, we changed the bar plot where the data were added to a scatter/line plot depicting the same information in a more compact display. Second, we moved the plots and description of Reward Bar and Trial Number encoding to the supplement, since these reiterate the same point made by the other data in the figure.

We have also included additional figures for MUA and summed MUA (Σ MUA), as requested, for Figures 6, 8, and 9. The analyses shown in Figure 7 and 10 were done for MUA, but cannot be done for Σ MUA, because they look at relationships between signals recorded from the same electrode. By definition Σ MUA aggregates across electrodes, so we can't perform this one-to-one mapping.

2. A very interesting extension could be to use the summed activity of not the entire neuronal data, but instead only some electrodes around the LFP electrode, and to quantify the area of the summed neuronal activity that matches the HGA activity. I think this would be a valuable addition that would significantly improve the impact of the paper.

We agree that this would be a very interesting analysis to conduct, however the anatomical precision of our data in this study is too limited to do exactly what the reviewer suggests. Here, our acute electrodes are scattered throughout OFC at irregular intervals, with ~4-8 placed in each hemisphere on a given day. We obtained systematic sampling by aggregating over sessions, but within session we have much more sparse sampling. For instance, we rarely had two electrodes < 3 mm apart in both AP and ML dimensions. In the Z direction, the fact that we placed these as depth electrodes into cortex with complex sulcal anatomy introduces minor errors in measurements of precisely how far apart two electrodes were. Ideally, to do this analysis justice, we would need data acquired simultaneously and at very regular spatial intervals, such as on a Utah array.

Despite these limitations, we attempted to address the spirit of the reviewer's idea, that is, to analyze a mass signal from an anatomical vicinity that is greater than 1 electrode but less than the entire recording field. We created partial hemisphere MUs by splitting the recording field in each hemisphere into anterior and posterior sectors (at AP 28) and adding spiking activity within each of these 4 areas for each session. Thus, each quadrant included ~5 mm x 6 mm as potential electrode locations. This resulted in MUs that included 2.3 ± 0.3 electrodes aggregated together (2.4 ± 0.4 in anterior and 2.2 ± 0.4 in posterior OFC; mean \pm 95% CI). We then carried out the same analyses to determine the encoding properties of these new MUs.

The partial-hemisphere MUs were almost indistinguishable from single electrode MUs. The figure in panel a below replicates the original bar plot showing the percent of each signal encoding each task variable (Neurons and HGA have been removed for space). Striped bars are single electrode MUA, hatched bars are total summed MUA, and gray bars are partial-hemisphere MUA. Overall, the gray bars follow closely with single electrode MUA encoding. Panels b-d show encoding of key variables over time, separated by encoding valence. In the top row is partial-hemisphere (PH) MUA, and in the bottom, single electrode MUA plots are reproduced from the manuscript. Again, all are very similar, with the exception that the n is smaller for PH-MUA.

Even when we aggregated over fairly large anatomical territories, we didn't find a clear change in MUA encoding by adding electrodes. We know from previous work that the area we included in each quadrant should be much larger than the area contributing to a HG signal, indicating that we can't make a strong statement about how large a territory of OFC is aggregated in HGA. These results are most likely due to the sparse sampling in this data set. Since only 2-3 electrodes recorded from each quadrant, we can't claim to actually be aggregating over the entire field, just sampling from it. Because of these shortcomings, we have excluded this analysis from the final manuscript.

3. Supplementary Figure 1c – HG activity appears to decrease before the picture onset and reaches a minimum at 0. Why is the HG higher before the onset? Do you see the same trend in the firing rates?

This pattern was the result of higher HGA following reward receipt and during inter-trial interval (ITI). There was approximately 2.5s between the end of a trial and the appearance of the fixation point that signaled the start of the next trial. Extending the time window back to 3s before picture onset (that is, overlapping with the reward epoch) showed that there was strong HGA at the end of the preceding trial, when the subject got a reward, that slowly decayed during the ITI. There was an evoked response at the time the fixation point appeared, with a continued decrease in HGA during the fixation period (First panels of the figure below; shading shows 99% CIs). This pattern was common across OFC, and the main difference between anterior and posterior electrodes was the early transient response immediately following the onset of a reward-predicting picture, that was found in anterior OFC, as we describe in the manuscript. (Note that in the

manuscript figures HGA was normalized to 500 ms during the fixation epoch, however for this plot we wanted to include the fixation epoch in the time period we are assessing, so we normalized to an epoch spanning -4s - to -3s).

Single neurons, on the other hand, had more heterogeneous responses. On average, there was not the same pattern of higher overall firing rates during the ITI (Middle panels). However, in agreement with HGA, the early transient response in anterior OFC following picture onset is weakly apparent among single units as well.

We also looked at MUA (Last panels), and found a pattern that began to approximate that seen in HGA, with slightly higher activity during the ITI that decreased to its lowest point just before the pictures appeared.

Therefore, to answer the reviewer's question more directly, the decreasing trend in HGA before picture onset is due to effects of the previous trial. We now include the following sentence in the legend for Supplementary Fig 1:

Note that the decrease in signal amplitude prior to picture onset resulted from a decrease over time following the reward epoch on the previous trial.

Since this is a fairly mundane explanation, we chose not to include the figures showing the effect.

4. Supplementary Figure 3 - can the authors also show firing rates as a function of AP as well?

A plot of firing rates is now included in this figure.

REVIEWERS' COMMENTS:

Reviewer #2 (Remarks to the Author):

Thank you for doing the additional analyses. I have no more concerns. I recommend the paper for publication.